# Fairness Behind a Veil of Ignorance:
# A Welfare Analysis for Automated Decision Making

**Hoda Heidari**
ETH Zürich
hheidari@inf.ethz.ch

**Claudio Ferrari**
ETH Zürich
ferraric@ethz.ch

**Krishna P. Gummadi**
MPI-SWS
gummadi@mpi-sws.org

**Andreas Krause**
ETH Zürich
krausea@ethz.ch

## Abstract

We draw attention to an important, yet largely overlooked aspect of evaluating fairness for automated decision making systems—namely risk and welfare considerations. Our proposed family of measures corresponds to the long-established formulations of cardinal social welfare in economics, and is justified by the Rawlsian conception of fairness *behind a veil of ignorance*. The convex formulation of our welfare-based measures of fairness allows us to integrate them as a constraint into any convex loss minimization pipeline. Our empirical analysis reveals interesting trade-offs between our proposal and (a) prediction accuracy, (b) group discrimination, and (c) Dwork *et al.*'s notion of individual fairness. Furthermore and perhaps most importantly, our work provides both heuristic justification and empirical evidence suggesting that a lower-bound on our measures often leads to bounded inequality in algorithmic outcomes; hence presenting the first computationally feasible mechanism for bounding individual-level inequality.

## 1 Introduction

Traditionally, data-driven decision making systems have been designed with the sole purpose of maximizing some system-wide measure of performance, such as accuracy or revenue. Today, these systems are increasingly employed to make consequential decisions for human subjects—examples include employment [Miller, 2015], credit lending [Petrasic *et al.*, 2017], policing [Rudin, 2013], and criminal justice [Barry-Jester *et al.*, 2015]. Decisions made in this fashion have long-lasting impact on people's lives and—absent a careful ethical analysis—may affect certain individuals or social groups negatively [Sweeney, 2013; Angwin *et al.*, 2016; Levin, 2016]. This realization has recently spawned an active area of research into quantifying and guaranteeing fairness for machine learning [Dwork *et al.*, 2012; Kleinberg *et al.*, 2017; Hardt *et al.*, 2016].

Virtually all existing formulations of algorithmic fairness focus on guaranteeing *equality* of some notion of *benefit* across different individuals or socially salient groups. For instance, demographic parity [Kamiran and Calders, 2009; Kamishima *et al.*, 2011; Feldman *et al.*, 2015] seeks to equalize the percentage of people receiving a particular outcome across different groups. Equality of opportunity [Hardt *et al.*, 2016] requires the equality of false positive/false negative rates. Individual fairness [Dwork *et al.*, 2012] demands that people who are equal with respect to the task at hand receive equal outcomes. In essence, the debate so far has mostly revolved around identifying the right notion of *benefit* and a tractable mathematical formulation for *equalizing* it.

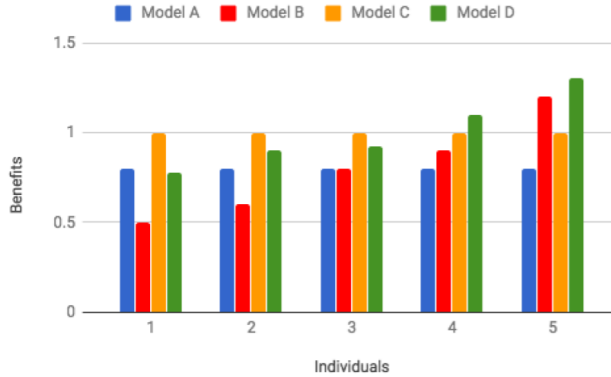

Figure 1: Predictive model A assigns the same benefit of 0.8 to everyone; model C assigns the same benefit of 1 to everyone; model B results in benefits $(0.5, 0.6, 0.8, 0.9, 1.2)$, and model D, $(0.78, 0.9, 0.92, 1.1, 1.3)$. Our proposed measures prefer A to B, C to D, and D to A.

The view of fairness as some form of equality is indeed an important perspective in the moral evaluation of algorithmic decision making systems—decision subjects often compare their outcomes with other similarly situated individuals, and these *interpersonal comparisons* play a key role in shaping their judgment of the system. We argue, however, that equality is not the only factor at play: we draw attention to two important, yet largely overlooked aspects of evaluating fairness of automated decision making systems—namely *risk* and *welfare*[1] considerations. The importance of these factors is perhaps best illustrated via a simple example.

**Example 1** *Suppose we have four decision making models A, B, C, D each resulting in a different benefit distribution across 5 groups/individuals $i_1, i_2, i_3, i_4, i_5$ (we will precisely define in Section 2 how benefits are computed, but for the time being and as a concrete example, suppose benefits are equivalent to salary predictions made through different regression models). Figure 1 illustrates the setting. Suppose a decision maker is tasked with determining which one of these alternatives is* ethically more desirable. *From an inequality minimizing perspective, A is clearly more desirable than B: note that both A, B result in the same total benefit of 4, and A distributes it equally across $i_1$, ..., $i_5$. With a similar reasoning, C is preferred to D. Notice, however, that by focusing on equality alone, A would be deemed more desirable than D, but there is an issue with this conclusion: almost everyone—expect for $i_1$ who sees a negligible drop of less than $2\%$ in their benefit—is significantly better off under D compared to A.[2] In other words, even though D results in unequal benefits and it does* not *Pareto-dominate A, collectively it results in higher* welfare *and lower* risk*, and therefore, both intuitively and from a* rational *point of view, it should be considered more desirable. With a similar reasoning, the decision maker should conclude C is more desirable than A, even though both provide benefits equally to all individuals.*

In light of this example and inspired by the long line of research on distributive justice in economics, in this paper we propose a natural family of measures for evaluating algorithmic fairness corresponding to the well-studied notions of *cardinal social welfare* in economics [Harsanyi, 1953, 1955]. Our proposed measures indeed prefer A to B, C to D, and D to A.

The interpretation of social welfare as a measure of fairness is justified by the concept of *veil of ignorance* (see [Freeman, 2016] for the philosophical background). Rawls [2009] proposes "veil of ignorance" as the ideal condition/mental state under which a policy maker can select the fairest among a number of political alternatives. He suggests that the policy maker performs the following thought experiment: imagine him/herself as an individual who knows nothing about the particular position they will be born in within the society, and is tasked with selecting the most just among a set of alternatives. According to the utilitarian doctrine in this hypothetical original/ex-ante position if the individual is *rational*, they would aim to minimize risk and insure against unlucky events in which they turn out to assume the position of a low-benefit individual. Note that decision making behind a

veil of ignorance is a purely imaginary condition: the decision maker can never in actuality be in this position, nonetheless, the thought experiment is useful in detaching him/her from the needs and wishes of a particular person/group, and consequently making a fair judgment. Our main conceptual contribution is to measure fairness in the context of algorithmic decision making by evaluating it from behind a veil of ignorance: our proposal is for the ML expert wishing to train a fair decision making model (e.g. to decide whether salary predictions are to be made using a neural network or a decision tree) to perform the aforementioned thought experiment: He/she should evaluate fairness of each alternative by taking the perspective of the algorithmic decision making subjects—but not any particular one of them: he/she must *imagine* themselves in a hypothetical setting where they know they will be born as one of the subjects, but don't know in advance which one. We consider the alternative he/she deems best behind this veil of ignorance to be the fairest.

To formalize the above, our core idea consists of comparing the expected *utility* a randomly chosen, *risk-averse* subject of algorithmic decision making receives under different predictive models. In the example above, if one is to choose between models A, D without knowing which one of the 5 individuals they will be, then the risk associated with alternative D is much less than that of A—under A the individual is going to receive a (relatively low) benefit of 0.8 with certainty, whereas under D with high probability (i.e. 4/5) they obtain a (relatively large) benefit of 0.9 or more, and with low probability (1/5) they receive a benefit of 0.78, roughly the same as the level of benefit they would attain under A. Such considerations of risk is precisely what our proposal seeks to quantify. We remark that in comparing two benefit distributions of the *same mean* (e.g. A, B or C, D in our earlier example), our risk-averse measures always prefer the more equal one (A is preferred to B and C is preferred to D). See Proposition 2 for the formal statement. Thus, our measures are inherently equality preferring. However, the key advantage of our measures of social welfare over those focusing on inequality manifests when, as we saw in the above example, comparing two benefit distributions of different means. In such conditions, inequality based measures are insufficient and may result in misleading conclusions, while risk-averse measures of social welfare are better suited to identify the fairest alternative. When comparing two benefit distributions of the same mean, social welfare and inequality would always yield identical conclusions.

Furthermore and from a computational perspective, our welfare-based measures of fairness are more convenient to work with due to their *convex* formulation. This allows us to integrate them as a constraint into any convex loss minimization pipeline, and solve the resulting problem efficiently and exactly. Our empirical analysis reveals interesting trade-offs between our proposal and (a) prediction accuracy, (b) group discrimination, and (c) Dwork *et al.*'s notion of individual fairness. In particular, we show how loss in accuracy increases with the degree of risk aversion, $\alpha$, and as the lower bound on social welfare, $\tau$, becomes more demanding. We observe that the difference between false positive/negative rates across different social groups consistently decreases with $\tau$. The impact of our constraints on demographic parity and Dwork *et al.*'s notion of individual fairness is slightly more nuanced and depends on the type of learning task at hand (regression vs. classification). Last but not least, we provide empirical evidence suggesting that a lower bound on social welfare often leads to bounded inequality in algorithmic outcomes; hence presenting the first computationally feasible mechanism for bounding individual-level inequality.

## 1.1 Related Work

Much of the existing work on algorithmic fairness has been devoted to the study of *discrimination* (also called *statistical-* or *group*-level fairness). Statistical notions require that given a classifier, a certain fairness metric is equal across all protected groups (see e.g. [Kleinberg *et al.*, 2017; Zafar *et al.*, 2017b,a]). Statistical notions of fairness fail to guarantee fairness at the individual level. Dwork *et al.* [2012] first formalized the notion of individual fairness for classification learning tasks, requiring that two individuals who are similar with respect to the task at hand receive similar classification outcomes. The formulation relies on the existence of a suitable similarity metric between individuals, and as pointed out by Speicher *et al.*, it does not take into account the variation in *social desirability* of various outcomes and people's merit for different decisions. Speicher *et al.* [2018] recently proposed a new measure for quantifying individual unfairness utilizing income *inequality indices* from economics and applying them to algorithmic benefit distributions. Both existing formulations of individual-level fairness focus solely on the *inter-personal* comparisons of algorithmic outcomes/benefits across individuals and do not account for *risk* and *welfare* considerations. Furthermore, we are not aware of computationally efficient mechanisms for bounding either of these notions.

We consider our family of measures to belong to the individual category: our welfare-based measures do *not* require knowledge of individuals' membership in protected groups, and compose the *individual level* utilities through *summation*. Note that Dwork *et al.* [2012] propose a stronger notion of individual fairness—one that requires a certain (minimum) condition to hold for every individual. As we will see shortly, a limiting case of our proposal (the limit of $\alpha = -\infty$) provides a similar guarantee in terms of benefits. While our main focus in this work is on individual-level fairness, our proposal can be readily extended to measure and constraint group-level unfairness.

Zafar *et al.* [2017c] recently proposed two preference-based notions of fairness at the group-level, called *preferred treatment* and *preferred impact*. A group-conditional classifier satisfies preferred treatment if no group collectively prefers another group's classifier to their own (in terms of average misclassification rate). This definition is based on the notion of *envy-freeness* [Varian, 1974] in economics and applies to group-conditional classifiers only. A classifier satisfies preferred impact if it Pareto-dominates an existing impact parity classifier (i.e. every group is better off using the former classifier compared to the latter). Pareto-dominance (to be defined precisely in Section 2) leads to a *partial* ordering among alternatives and usually in practice, does not have much bite (recall, for instance, the comparison between models A, D in our earlier example). Similar to [Zafar *et al.*, 2017c], our work can be thought of as a preference-based notions of fairness, but unlike their proposal our measures lead to a *total* ordering among all alternatives, and can be utilized to measure both individual and group-level (un)fairness.

Further discussion of related work can be found in Appendix A.

## 2 Our Proposed Family of Measures

We consider the standard supervised learning setting: A learning algorithm receives the training data set $D = \{(\mathbf{x}_i, y_i)\}_{i=1}^n$ consisting of $n$ instances, where $\mathbf{x}_i \in \mathcal{X}$ specifies the feature vector for individual $i$ and $y_i \in \mathcal{Y}$, the *ground truth* label for him/her. The training data is sampled i.i.d. from a distribution $P$ on $\mathcal{X} \times \mathcal{Y}$. Unless specified otherwise, we assume $\mathcal{X} \subseteq \mathbb{R}^k$, where $k$ denotes the number of features. To avoid introducing extra notation for an intercept, we assume feature vectors are in homogeneous form, i.e. the $k$th feature value is 1 for every instance. The goal of a learning algorithm is to use the training data to fit a *model* (or hypothesis) $h : \mathcal{X} \to \mathcal{Y}$ that accurately predicts the label for new instances. Let $\mathcal{H}$ be the hypothesis class consisting of all the models the learning algorithm can choose from. A learning algorithm receives $D$ as the input; then utilizes the data to select a model $h \in \mathcal{H}$ that minimizes some notion of loss, $L_D(h)$. When $h$ is clear from the context, we use $\hat{y}_i$ to refer to $h(\mathbf{x}_i)$.

We assume there exists a benefit function $b : \mathcal{Y} \times \mathcal{Y} \to \mathbb{R}$ that quantifies the benefit an individual with ground truth label $y$ receives, if the model predicts label $\hat{y}$ for them.[3] The benefit function is meant to capture the *signed discrepancy* between an individual's predicted outcome and their true/deserved outcome. Throughout, for simplicity we assume higher values of $\hat{y}$ correspond to more desirable outcomes (e.g. loan or salary amount). With this assumption in place, a benefit function must assign a high value to an individual if their predicted label is greater (better) than their deserved label, and a low value if an individual receives a predicted label less (worse) than their deserved label. The following are a few examples of benefit functions that satisfy this: $b(y, \hat{y}) = \hat{y} - y$; $b(y, \hat{y}) = log\left(1 + e^{\hat{y}-y}\right)$; $b(y, \hat{y}) = \hat{y}/y$.

In order to maintain the convexity of our fairness constraints, throughout this work, we will focus on benefit functions that are positive and linear in $\hat{y}$. In general (e.g. when the prediction task is regression or multi-class classification) this limits the benefit landscape that can be expressed, but in the important special case of binary classification, the following Proposition establishes that this restriction is without loss of generality[4]. That is, we can attach an arbitrary combination of benefit values to the four possible $(y, \hat{y})$-pairs (i.e. false positives, false negatives, true positives, true negative).

**Proposition 1** *For $y, \hat{y} \in \{0, 1\}$, let $\bar{b}_{y,\hat{y}} \in \mathbb{R}$ be arbitrary constants specifying the benefit an individual with ground truth label $y$ receives when their predicted label is $\hat{y}$. There exists a linear benefit function of form $c_y \hat{y} + d_y$ such that for all $y, \hat{y} \in \{0, 1\}$, $b(y, \hat{y}) = \bar{b}_{y,\hat{y}}$.*

In order for $\bar{b}$'s in the above proposition to reflect the signed discrepancy between $y$ and $\hat{y}$, it must hold that $\bar{b}_{1,0} < \bar{b}_{0,0} \leq \bar{b}_{1,1} < \bar{b}_{0,1}$. Given a model $h$, we can compute its corresponding benefit profile $\mathbf{b} = (b_1, \cdots, b_n)$ where $b_i$ denotes individual $i$'s benefit: $b_i = b(y_i, \hat{y}_i)$. A benefit profile $\mathbf{b}$ *Pareto-dominates* $\mathbf{b}'$ (or in short $\mathbf{b} \succeq \mathbf{b}'$), if for all $i = 1, \cdots, n$, $b_i \geq b_i'$.

Following the economic models of risk attitude, we assume the existence of a utility function $u : \mathbb{R} \rightarrow \mathbb{R}$, where $u(b)$ represent the utility derived from algorithmic benefit $b$. We will focus on *Constant Relative Risk Aversion (CRRA)* utility functions. In particular, we take $u(b) = b^\alpha$ where $\alpha = 1$ corresponds to risk-neutral, $\alpha > 1$ corresponds to risk-seeking, and $0 \leq \alpha < 1$ corresponds to risk-averse preferences. Our main focus in this work is on values of $0 < \alpha < 1$: the larger one's initial benefit is, the smaller the added utility he/she derives from an increase in his/her benefit. While in principle our model can allow for different risk parameters for different individuals ($\alpha_i$ for individual $i$), for simplicity throughout we assume all individuals have the same risk parameter. Our measures assess the fairness of a decision making model via the expected *utility* a randomly chosen, *risk-averse* individual receives as the result of being subject to decision making through that model. Formally, our measure is defined as follows: $\mathcal{U}_P(h) = \mathbb{E}_{(\mathbf{x}_i, y_i) \sim P}[u(b(y_i, h(\mathbf{x}_i)))]$. We estimate this expectation by $\mathcal{U}_D(h) = \frac{1}{n}\sum_{i=1}^{n} u(b(y_i, h(\mathbf{x}_i)))$.

**Connection to Cardinal Welfare**   Our proposed family of measures corresponds to a particular subset of cardinal social welfare functions. At a high level, a cardinal social welfare function is meant to rank different distributions of welfare across individuals, as more or less desirable in terms of distributive justice [Moulin, 2004]. More precisely, let $\mathcal{W}$ be a welfare function defined over benefit vectors, such that given any two benefit vectors $\mathbf{b}$ and $\mathbf{b}'$, $\mathbf{b}$ is considered more desirable than $\mathbf{b}'$ if and only if $\mathcal{W}(\mathbf{b}) \geq \mathcal{W}(\mathbf{b}')$. The rich body of work on welfare economics offers several axioms to characterize the set of all welfare functions that pertain to collective rationality or fairness. Any such function, $\mathcal{W}$, must satisfy the following axioms [Sen, 1977; Roberts, 1980]:

1. **Monotonicity:** If $\mathbf{b}' \succ \mathbf{b}$, then $\mathcal{W}(\mathbf{b}') > \mathcal{W}(\mathbf{b})$. That is, if everyone is better off under $\mathbf{b}'$, then $\mathcal{W}$ should strictly prefer it to $\mathbf{b}$.

2. **Symmetry:** $\mathcal{W}(b_1, \ldots, b_n) = \mathcal{W}(b_{(1)}, \cdots, b_{(n)})$. That is, $\mathcal{W}$ does not depend on the identity of the individuals, but only their benefit levels.

3. **Independence of unconcerned agents:** $\mathcal{W}$ should be independent of individuals whose benefits remain at the same level. Formally, let $(\mathbf{b}|^i a)$ be a benefit vector that is identical to $\mathbf{b}$, expect for the $i$th component which has been replaced by $a$. The property requires that for all $\mathbf{b}, \mathbf{b}', a, c$, $\mathcal{W}(\mathbf{b}|^i a) \leq \mathcal{W}(\mathbf{b}'|^i a) \Leftrightarrow \mathcal{W}(\mathbf{b}|^i c) \leq \mathcal{W}(\mathbf{b}'|^i c)$.

It has been shown that every *continuous*[5] social welfare function $\mathcal{W}$ with properties 1–3 is additive and can be represented as $\sum_{i=1}^{n} w(b_i)$. According to the Debreu-Gorman theorem [Debreu, 1959; Gorman, 1968], if in addition to 1–3, $\mathcal{W}$ satisfies:

4. **Independence of common scale:** For any $c > 0$, $\mathcal{W}(\mathbf{b}) \geq \mathcal{W}(\mathbf{b}') \Leftrightarrow \mathcal{W}(c\mathbf{b}) \geq \mathcal{W}(c\mathbf{b}')$. The simultaneous rescaling of every individual benefit, should not affect the relative order of $\mathbf{b}, \mathbf{b}'$.

then it belongs to the following one-parameter family: $\mathcal{W}_\alpha(b_1, \ldots, b_n) = \sum_{i=1}^{n} w_\alpha(b_i)$, where (a) for $\alpha > 0$, $w_\alpha(b) = b^\alpha$; (b) for $\alpha = 0$, $w_\alpha(b) = \ln(b)$; and (c) for $\alpha < 0$, $w_\alpha(b) = -b^\alpha$. Note that the limiting case of $\alpha \rightarrow -\infty$ is equivalent to the leximin ordering (or Rawlsian max-min welfare).

Our focus in this work is on $0 < \alpha < 1$. In this setting, our measures exhibit aversion to pure inequality. More precisely, they satisfy the following important property:

5. **Pigou-Dalton transfer principle [Pigou, 1912; Dalton, 1920]:** Transferring benefit from a high-benefit to a low-benefit individual must increase social welfare, that is, for any $1 \leq i < j \leq n$ and $0 < \delta < \frac{b_{(j)} - b_{(i)}}{2}$, $\mathcal{W}(b_{(1)}, \cdots, b_{(i)} + \delta, \cdots, b_{(j)} - \delta, \cdots, b_{(n)}) > \mathcal{W}(\mathbf{b})$.

## 2.1 Our In-processing Method to Guarantee Fairness

To guarantee fairness, we propose minimizing loss subject to a lower bound on our measure:

$$\min_{h \in \mathcal{H}} \quad L_D(h)$$
$$\text{s.t.} \quad \mathcal{U}_D(h) \geq \tau$$

where the parameter $\tau$ specifies a lower bound that must be picked carefully to achieve the right tradeoff between accuracy and fairness. As a concrete example, when the learning task is linear regression, $b(y, \hat{y}) = \hat{y} - y + 1$, and the degree of risk aversion in $\alpha$, this optimization amounts to:

$$\min_{\boldsymbol{\theta} \in \mathcal{H}} \quad \sum_{i=1}^{n} (\boldsymbol{\theta}.\mathbf{x}_i - y_i)^2$$
$$\text{s.t.} \quad \sum_{i=1}^{n} (\boldsymbol{\theta}.\mathbf{x}_i - y_i + 1)^{\alpha} \geq \tau n \tag{1}$$

Note that both the objective function and the constraint in (1) are convex in $\boldsymbol{\theta}$, therefore, the optimization can be solved efficiently and exactly.

**Connection to Inequality Measures** Speicher *et al.* [2018] recently proposed quantifying individual-level unfairness utilizing a particular inequality index, called generalized entropy. This measure satisfies four important axioms: symmetry, population invariance, 0-normalization[6], and the Pigou–Dalton transfer principle. Our measures satisfy all the aforementioned axioms, except for 0-normalization. Additionally and in contrast with measures of inequality—where the goal is to capture interpersonal comparison of benefits—our measure is monotone and independent of unconcerned agents. The latter two are the fundamental properties that set our proposal apart from measures of inequality.

Despite these fundamental differences, we will shortly observe in Section 3 that lower-bounding our measures often in practice leads to low inequality. Proposition 2 provides a heuristic explanation for this: Imposing a lower bound on social welfare is equivalent to imposing an upper bound on inequality if we restrict attention to the region where benefit vectors are all of the same mean. More precisely, for a fixed mean benefit value, our proposed measure of fairness results in the same total ordering as the Atkinson's index [Atkinson, 1970]. The index is defined as follows:

$$A_\beta(b_1, \ldots, b_n) = \begin{cases} 1 - \frac{1}{\mu} \left( \frac{1}{n} \sum_{i=1}^{n} b_i^{1-\beta} \right)^{1/(1-\beta)} & \text{for } 0 \leq \beta \neq 1 \\ 1 - \frac{1}{\mu} \left( \prod_{i=1}^{n} b_i \right)^{1/n} & \text{for } \beta = 1, \end{cases}$$

where $\mu = \frac{1}{n} \sum_{i=1}^{n} b_i$ is the mean benefit. Atkinson's inequality index is a *welfare*-based measure of inequality: The measure compares the actual average benefit individuals receive under benefit distribution $\mathbf{b}$ (i.e. $\mu$) with its Equally Distributed Equivalent (EDE)—the level of benefit that if obtained by every individual, would result in the same level of welfare as that of $\mathbf{b}$ (i.e. $\frac{1}{n} \sum_{i=1}^{n} b_i^{1-\beta}$). It is easy to verify that for $0 < \alpha < 1$, the generalized entropy and Atkinson index result in the same total ordering among benefit distributions (see Proposition 3 in Appendix B). Furthermore, for a fixed mean benefit $\mu$, our measure results in the same indifference curves and total ordering as the Atkinson index with $\beta = 1 - \alpha$.

**Proposition 2** *Consider two benefit vectors* $\mathbf{b}, \mathbf{b}' \succ \mathbf{0}$ *with equal means* $(\mu = \mu')$. *For* $0 < \alpha < 1$, $A_{1-\alpha}(\mathbf{b}) \geq A_{1-\alpha}(\mathbf{b}')$ *if and only if* $\mathcal{W}_\alpha(\mathbf{b}) \leq \mathcal{W}_\alpha(\mathbf{b}')$.

**Tradeoffs Among Different Notions of Fairness** We end this section by establishing the existence of multilateral tradeoffs among social welfare, accuracy, individual, and statistical notions of fairness. We illustrate this by finding the predictive model that optimizes each of these quantities. In Table 1 we compare these optimal predictors in two different cases: 1) In the *realizable* case, we assume the existence of a hypothesis $h^* \in \mathcal{H}$ such that $y = h^*(\mathbf{x})$, i.e., $h^*$ achieves perfect prediction accuracy. 2) In the *unrealizable* case, we assume the existence of a hypothesis $h^* \in \mathcal{H}$, such that $h^*(\mathbf{x}) = \mathbb{E}[y|\mathbf{x}]$), i.e., $h^*$ is the Bayes Optimal Predictor. We use the following notations:

Table 1: Optimal predictions with respect to different fairness notions.

| | Classification | | Regression | |
|---|---|---|---|---|
| | Realizable | Unrealizable | Realizable | Unrealizable |
| Social welfare | $\hat{y} \equiv 1$ | $\hat{y} \equiv 1$ | $\hat{y} \equiv y_{\max}$ | $\hat{y} \equiv y_{\max}$ |
| Atkinson index | $\hat{y} = h^*(\mathbf{x})$ | $\hat{y} \equiv 1$ | $\hat{y} = h^*(\mathbf{x})$ | $\hat{y} \equiv y_{\max}$ |
| Dwork et al.'s notion | $\hat{y} \equiv 0$ or $1$ | $\hat{y} \equiv 1$ or $0$ | $\hat{y} \equiv c$ | $\hat{y} \equiv c$ |
| Mean difference | $\hat{y} \equiv 0$ or $1$ | $\hat{y} \equiv 1$ or $0$ | $\hat{y} \equiv c$ | $\hat{y} \equiv c$ |
| Positive residual diff. | $\hat{y} \equiv 0$ or $\hat{y} = h^*(\mathbf{x})$ | $\hat{y} \equiv 0$ | $\hat{y} \equiv y_{\min}$ or $\hat{y} = h^*(\mathbf{x})$ | $\hat{y} \equiv y_{\min}$ |
| Negative residual diff. | $\hat{y} \equiv 1$ or $\hat{y} = h^*(\mathbf{x})$ | $\hat{y} \equiv 1$ | $\hat{y} \equiv y_{\max}$ or $\hat{y} = h^*(\mathbf{x})$ | $\hat{y} \equiv y_{\max}$ |

$y_{\max} = \max_{h \in \mathcal{H}, \mathbf{x} \in \mathcal{X}} h(\mathbf{x})$ and $y_{\min} = \min_{h \in \mathcal{H}, \mathbf{x} \in \mathcal{X}} h(\mathbf{x})$. The precise definition of each notion in Table 1 can be found in Appendix C.

As illustrated in Table 1, there is no unique predictors that simultaneously optimizes social welfare, accuracy, individual, and statistical notions of fairness. Take the unrealizable classification as an example. Optimizing for accuracy requires the predictions to follow the Bayes optimal classifier. A lower bound on social welfare requires the model to predict the desirable outcome (i.e. 1) for a large fraction of the population. To guarantee low positive residual difference, all individuals must be predicted to belong to the negative class. In the next Section, we will investigate these tradeoffs in more detail and through experiments on two real-world datasets.

## 3    Experiments

In this section, we empirically illustrate our proposal, and investigate the tradeoff between our family of measures and accuracy, as well as existing definitions of group discrimination and individual fairness. We ran our experiments on a classification data set (Propublica's *COMPAS dataset* [Larson *et al.*, 2016]), as well as a regression dataset (*Crime and Communities data set* [Lichman, 2013]).[7] For regression, we defined the benefit function as follows: $b(y, \hat{y}) = \hat{y} - y + 1$. On the Crime data set this results in benefit levels between 0 and 2. For classification, we defined the benefit function as follows: $b(y, \hat{y}) = c_y \hat{y} + d_y$ where $y \in \{-1, 1\}$, $c_1 = 0.5$, $d_1 = 0.5$, and $c_{-1} = 0.25$, $d_{-1} = 1.25$. This results in benefit levels 0 for false negatives, 1 for true positives and true negatives, and 1.5 for false positives.

**Welfare as a Measure of Fairness**    Our proposed family of measures is *relative* by design: It allows for meaningful comparison among different unfair alternatives. Furthermore, there is no unique value of our measures that always correspond to perfect fairness. This is in contrast with previously proposed, *absolute* notions of fairness which characterize the *condition* of *perfect* fairness—as opposed to measuring the degree of unfairness of various unfair alternatives. We start our empirical analysis by illustrating that our proposed measures can compare and rank different predictive models. We trained the following models on the COMPAS dataset: a multi-layered perceptron, fully connected with one hidden layer with 100 units (NN), the AdaBoost classifier (Ada), Logistic Regression (LR), a decision tree classifier (Tree), a nearest neighbor classifier (KNN). Figure 2 illustrates how these learning models compare with one another according to accuracy, Atkinson index, and social welfare. All values were computed using 20-fold cross validation. The confidence intervals are formed assuming samples come from Student's t distribution. As shown in Figure 2, the rankings obtained from Atkinson index and social welfare are identical. Note that this is consistent with Proposition 2. Given the fact that all models result in similar mean benefits, we expect the rankings to be consistent.

**Impact on Model Parameters**    Next, we study the impact of changing $\tau$ on the trained model parameters (see Figure 3a). We observe that as $\tau$ increases, the intercept continually rises to guarantee high levels of benefit and social welfare. On the COMPAS dataset, we notice an interesting trend for the binary feature sex (0 is female, 1 is male); initially being male has a negative weight and thus a negative impact on the classification outcome, but as $\tau$ is increased, the sign changes to positive to ensure men also get high benefits. The trade-offs between our proposed measure and prediction

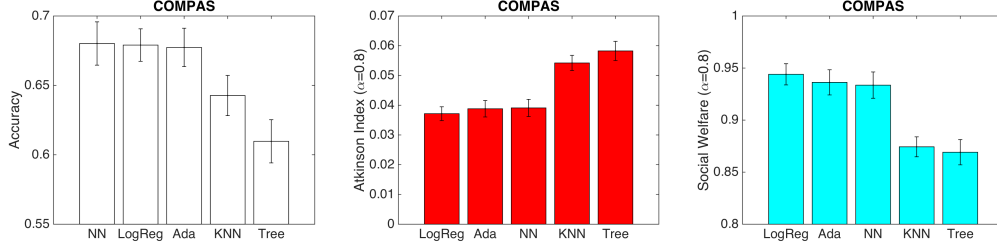

Figure 2: Comparison of different learning models according to accuracy, social welfare ($\alpha = 0.8$) and Atkinson index ($\beta = 0.2$). The mean benefits are 0.97 for LogReg, 0.96 for NN, 0.96 for AdaBoost, 0.89 for KNN, and 0.89 for Tree. Note that for Atkinson measure, smaller values correspond to fairer outcomes, where as for social welfare larger values reflect greater fairness.

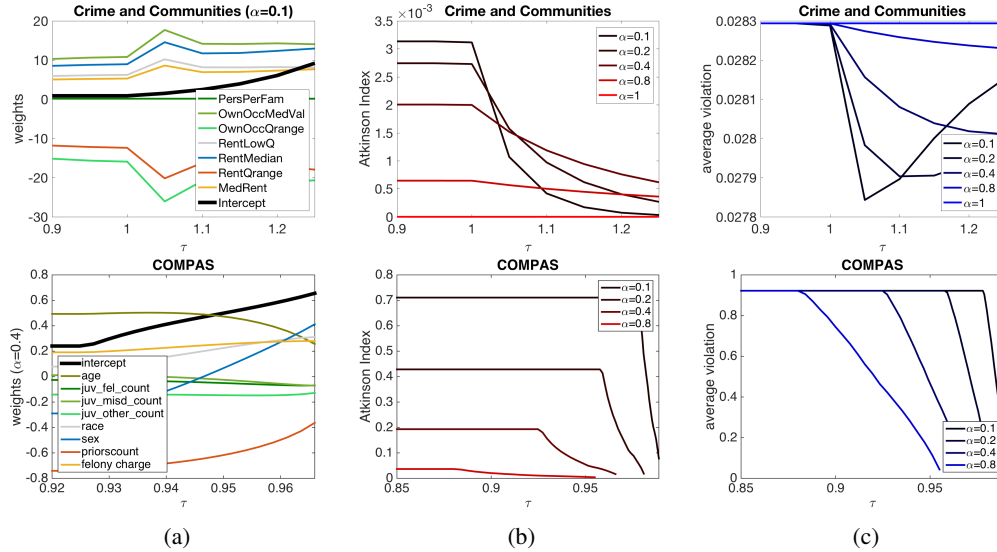

Figure 3: (a) Changes in weights—$\boldsymbol{\theta}$ in linear and logistic regression—as the function of $\tau$. Note the continuous rise of the intercept with $\tau$. (b) Atkinson index as a function of the threshold $\tau$. Note the consistent decline in inequality as $\tau$ increases. (c) Average violation of Dwork *et al.*'s constraints as a function of $\tau$. Trends are different for regression and classification.

accuracy can be found in Figure 5 in Appendix C. As one may expect, imposing more restrictive fairness constraints (larger $\tau$ and smaller $\alpha$), results in higher loss of accuracy.

Next, we will empirically investigate the tradeoff between our family of measures and existing definitions of group discrimination and individual fairness. Note that since our proposed family of measures is relative, we believe it is more suitable to focus on tradeoffs as opposed to *impossibility* results. (Existing impossibility results (e.g. [Kleinberg *et al.*, 2017]) establish that a number of absolute notions of fairness cannot hold simultaneously.)

**Trade-offs with Individual Notions**  Figures 3b, 3c illustrate the impact of bounding our measure on existing individual measures of fairness. As expected, we observe that higher values of $\tau$ (i.e. social welfare) consistently result in lower inequality. Note that for classification, $\tau$ cannot be arbitrarily large (due to the infeasibility of achieving arbitrarily large social welfare levels). Also as expected, smaller $\alpha$ values (i.e. higher degrees of risk aversion) lead to a faster drop in inequality. The impact of our mechanism on the average violation of Dwork *et al.*'s constraints is slightly more nuanced: as $\tau$ increases, initially the average violation of Dwork *et al.*'s pairwise constraints go down. For classification, the decline continues until the measure reaches 0—which is what we expect the measure to amount to once almost every individual receives the positive label. For regression in contrast, the initial decline is followed by a phase in which the measure quickly climbs back up

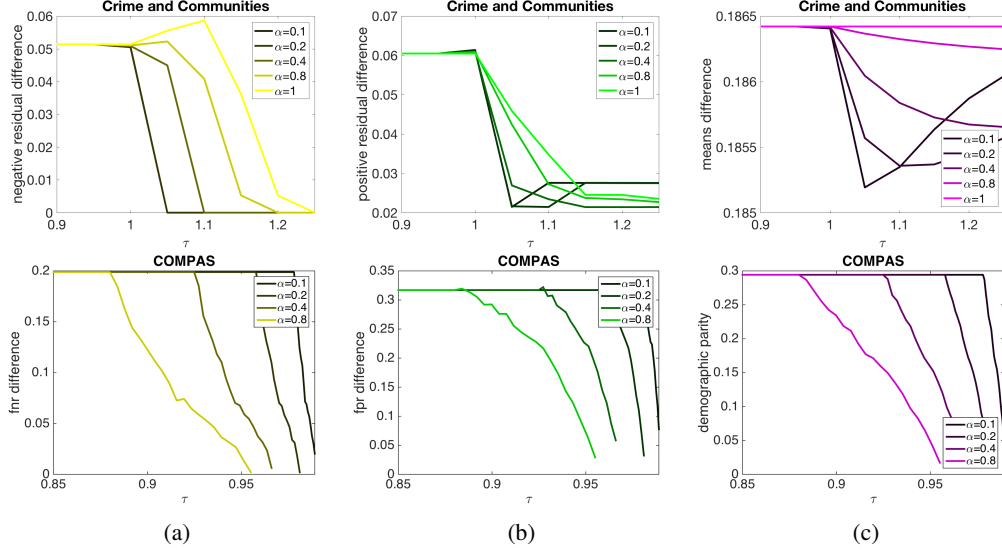

Figure 4: Group discrimination as a function of $\tau$ for different values of $\alpha$. (a) Negative residual difference is decreasing with $\tau$ and approaches $0$. (b) Positive residual difference monotonically approaches a certain asymptote. (c) Note the striking similarity of patterns for the average violation of Dwork *et al.*'s constraints and mean difference.

to its initial (high) value. The reason is for larger values of $\tau$, the high level of social welfare is achieved mainly by means of adding a large intercept to the unconstrained model's predictions (see Figure 3a). Due to its translation invariance property, the addition of an intercept cannot limit the average violation of Dwork *et al.*'s constraints.

**Trade-offs with Statistical Notions**    Next, we illustrate the impact of bounding our measure on statistical measures of fairness. For the Crime and Communities dataset, we assumed a neighborhood belongs to the protected group if and only if the majority of its residents are non-Caucasian, that is, the percentage of African-American, Hispanic, and Asian residents of the neighborhood combined, is above $50\%$. For the COMPAS dataset we took race as the sensitive feature. Figure 4a shows the impact of $\tau$ and $\alpha$ on false negative rate difference and its continuous counterpart, negative residual difference. As expected, both quantities decrease with $\tau$ until they reach $0$—when everyone receives a label at least as large as their ground truth. The trends are similar for false positive rate difference and its continuous counterpart, positive residual difference (Figure 4b). Note that in contrast to classification, on our regression data set, even though positive residual difference decreases with $\tau$, it never reaches $0$. Figure 4c shows the impact of $\tau$ and $\alpha$ on demographic parity and its continuous counterpart, means difference. Note the striking similarity between this plot and Figure 3c. Again here for large values of $\tau$, guaranteeing high social welfare requires adding a large intercept to the unconstrained model's prediction. See Proposition 4 in Appendix B, where we formally prove this point for the special case of linear predictors. The addition of intercept in this fashion, cannot put an upper-bound on a translation-invariant measure like mean difference.

## 4    Summary and Future Directions

Our work makes an important connection between the growing literature on fairness for machine learning, and the long-established formulations of cardinal social welfare in economics. Thanks to their convexity, our measures can be bounded as part of any convex loss minimization program. We provided evidence suggesting that constraining our measures often leads to bounded inequality in algorithmic outcomes. Our focus in this work was on a *normative* theory of how *rational* individuals should compare different algorithmic alternatives. We plan to extend our framework to *descriptive* behavioural theories, such as prospect theory [Kahneman and Tversky, 2013], to explore the *human perception of fairness* and contrast it with normative prescriptions.

## Acknowledgments

H. Heidari and A. Krause acknowledge support from CTI grant no. 27248.1 PFES-ES. Krishna P. Gummadi was supported in part by a European Research Council (ERC) Advanced Grant "Foundations for Fair Social Computing" (No. 789373).

## Footnotes

[1]We define *welfare* precisely in Sec. 2, but for now it can be taken as the sum of benefits across all subjects.

[2]In political philosophy, this problem is sometimes referred to as the "leveling down objection to equality".

[3]Our formulation allows the benefit function to depend on $\mathbf{x}$ and other available information about the individual. As long the formulation is linear in the predicted label $\hat{y}$, our approach remains computationally efficient. For simplicity and ease of interpretation, however, we focus on benefit functions that depend on $y$ and $\hat{y}$, only.

[4]All proofs can be found in Appendix B.

[5]That is, for every vector $\mathbf{b}$, the set of vectors weakly better than $\mathbf{b}$ (i.e. $\{\mathbf{b}' : \mathbf{b}' \succeq \mathbf{b}\}$) and the set of vectors weakly worse than $\mathbf{b}$ (i.e. $\{\mathbf{b}' : \mathbf{b}' \preceq \mathbf{b}\}$) are closed sets.

[6]0-normalization requires the inequality index to be 0 if and only if the distribution is perfectly equal/uniform.

[7]A more detailed description of the data sets and our preprocessing steps can be found in Appendix C.

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
