[Supplementary Material · neurips_2018_long.pdf]

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

# A  Related Work (Continued)

Also related to our work is [Corbett-Davies *et al.*, 2017], where authors propose maximizing an objective called "immediate utility" while satisfying existing fairness constraints. Immediate utility is meant to capture the impact of a decision rule on the society (e.g. on public safety when the task is to predict recidivism), and is composed of two terms: the expected number of true positives (e.g. number of crimes prevented), and the expected cost of positive labels (e.g. cost of detention). Note that our proposal is conceptually different from immediate utility in that we are concerned with the *individual-level* utility—i.e. the utility an individual obtains as the result of being *subject* to algorithmic decision making—whereas immediate utility is concerned with the impact of decisions on the society. For example, while it might be beneficial from the perspective of a high-risk defendant to be released, the societal cost of releasing him/her into the community is regarded as high. Furthermore and from a normative perspective, immediate utility is proposed as a replacement for prediction accuracy, whereas our measures are meant to capture desirability of algorithmic outcomes from the perspective of individuals subject to it.

Several papers in economics have studied the relationship between inequality aversion and risk aversion [Schwartz and Winship, 1980; Dagum, 1990]. At a high level, it is widely understood that the larger the relative risk aversion is, the more an individual choosing between different societies behind a "veil of ignorance" will be willing to trade-off expected benefit in order to achieve a more equal distribution. The following papers attempt to further clarify the link between evaluating risk *ex-ante* and evaluating inequality *ex-post*: Cowell and Schokkaert [2001] and Carlsson *et al.* [2005] empirically measure individuals' perceptions and preferences for risk and inequality through human-subject experiments. Amiel and Cowell [2003] establish a general relationship between the standard form of the social-welfare function and the "reduced-form" version that is expressed in terms of inequality and mean income.

# B  Omitted Technical Material

**Proof of Proposition 1**    Solving the following system of equations,

$$\forall y, \hat{y} \in \{0,1\} : c_y \hat{y} + d_y = \bar{b}_{y,\hat{y}}$$

we obtain: $c_0 = \bar{b}_{0,1} - \bar{b}_{0,0}$, $c_1 = \bar{b}_{1,1} - \bar{b}_{1,0}$, $d_0 = \bar{b}_{0,0}$, and $d_1 = \bar{b}_{1,0}$. ∎

**Proof of Proposition 2**    We have that:

$$
\begin{aligned}
A_{1-\alpha}(\mathbf{b}) \geq A_{1-\alpha}(\mathbf{b}') \quad &\Rightarrow \quad 1 - \frac{1}{\mu}\left(\frac{1}{n}\sum_{i=1}^{n} b_i^{\alpha}\right)^{1/\alpha} \geq 1 - \frac{1}{\mu'}\left(\frac{1}{n}\sum_{i=1}^{n} b_i'^{\alpha}\right)^{1/\alpha} \\
&\Leftrightarrow \quad \frac{1}{\mu}\left(\frac{1}{n}\sum_{i=1}^{n} b_i^{\alpha}\right)^{1/\alpha} \leq \frac{1}{\mu'}\left(\frac{1}{n}\sum_{i=1}^{n} b_i'^{\alpha}\right)^{1/\alpha} \\
&\Leftrightarrow \quad \left(\frac{1}{n}\sum_{i=1}^{n} b_i^{\alpha}\right)^{1/\alpha} \leq \left(\frac{1}{n}\sum_{i=1}^{n} b_i'^{\alpha}\right)^{1/\alpha} \\
&\Leftrightarrow \quad \sum_{i=1}^{n} b_i^{\alpha} \leq \sum_{i=1}^{n} b_i'^{\alpha} \\
&\Leftrightarrow \quad \mathcal{W}_{\alpha}(\mathbf{b}) \leq \mathcal{W}_{\alpha}(\mathbf{b}')
\end{aligned}
$$

∎

**Generalized entropy vs. Atkinson index**    Let $\mathcal{G}_{\alpha}(\mathbf{b})$ specify the generalized entropy, where

$$\mathcal{G}_{\alpha}(\mathbf{b}) = \frac{1}{n\alpha(\alpha-1)} \sum_{i=1}^{n}\left[\left(\frac{b_i}{\mu}\right)^{\alpha} - 1\right]$$

**Proposition 3** *Suppose $0 < \alpha < 1$. For any two benefit distributions $\mathbf{b}, \mathbf{b}'$, $\mathcal{A}_{1-\alpha}(\mathbf{b}) \geq \mathcal{A}_{1-\alpha}(\mathbf{b}')$ if and only if $\mathcal{G}_\alpha(\mathbf{b}) \geq \mathcal{G}_\alpha(\mathbf{b}')$.*

**Proof** First note that for any distribution $\mathbf{b}$, $\mathcal{A}_{1-\alpha}(\mathbf{b}) = 1 - (\alpha(\alpha - 1)\mathcal{G}_\alpha(\mathbf{b}) + 1)^{1/\alpha}$. We have that

$$
\begin{aligned}
\mathcal{A}_{1-\alpha}(\mathbf{b}) \geq \mathcal{A}_{1-\alpha}(\mathbf{b}') \quad &\Leftrightarrow \quad 1 - \mathcal{A}_{1-\alpha}(\mathbf{b}) \leq 1 - \mathcal{A}_{1-\alpha}(\mathbf{b}') \\
&\Leftrightarrow \quad \alpha \ln\left(1 - \mathcal{A}_{1-\alpha}(\mathbf{b})\right) \leq \alpha \ln\left(1 - \mathcal{A}_{1-\alpha}(\mathbf{b}')\right) \\
&\Leftrightarrow \quad \ln\left(\alpha(\alpha - 1)\mathcal{G}_\alpha(\mathbf{b}) + 1\right) \leq \ln\left(\alpha(\alpha - 1)\mathcal{G}_\alpha(\mathbf{b}') + 1\right) \\
&\Leftrightarrow \quad \alpha(\alpha - 1)\mathcal{G}_\alpha(\mathbf{b}) + 1 \leq \alpha(\alpha - 1)\mathcal{G}_\alpha(\mathbf{b}') + 1 \\
&\Leftrightarrow \quad \mathcal{G}_\alpha(\mathbf{b}) \geq \mathcal{G}_\alpha(\mathbf{b}')
\end{aligned}
$$

∎

**The role of intercept in guaranteeing high social welfare** Consider the problem of minimizing mean squared error subject to fairness constraints. We observed empirically that for large values of $\tau$, guaranteeing high social welfare requires adding a large intercept to the unconstrained model's prediction. This does not, however, put a limit on the mean difference and Dwork's measure. Next, we formally prove this point for the special case in which labels are all a linear function of the feature vectors.

**Proposition 4** *Suppose there exists a weight vector $\boldsymbol{\theta}^*$ such that for all $(\mathbf{x}_i, y_i) \in D$, $y_i = \boldsymbol{\theta}^* . \mathbf{x}_i$. Then for any $0 < \alpha < 1$ and $\tau > 1$, the optimal solution to (1) is $\boldsymbol{\theta}' = \boldsymbol{\theta}^* + \tau' \mathbf{e}_k$, where $\tau' = \tau^{1/\alpha} - 1$.*

**Proof** Given that Slater's condition trivially holds, we verify the optimality of $\boldsymbol{\theta}^* + \tau' \mathbf{e}_k$, along with dual multiplier

$$
\lambda' = \frac{2}{\alpha \tau} \tau^{1/\alpha}(\tau^{1/\alpha} - 1),
$$

using KKT conditions:

- **Stationarity** requires that:

$$
\sum_{i=1}^{n} 2\mathbf{x}_i(\boldsymbol{\theta}' . \mathbf{x}_i - y_i) = \lambda' \alpha \sum_{i=1}^{n} \mathbf{x}_i(\boldsymbol{\theta}' . \mathbf{x}_i - y_i + 1)^{\alpha - 1}.
$$

This is equivalent to

$$
\begin{aligned}
2\tau' \sum_{i=1}^{n} \mathbf{x}_i &= \lambda' \alpha \tau^{(\alpha-1)/\alpha} \sum_{i=1}^{n} \mathbf{x}_i \\
\Leftrightarrow \quad 2(\tau^{1/\alpha} - 1) &= \lambda' \alpha \tau^{1 - 1/\alpha} \\
\Leftrightarrow \quad \frac{2}{\alpha \tau} \tau^{1/\alpha}(\tau^{1/\alpha} - 1) &= \lambda'
\end{aligned}
$$

- **Dual feasibility** requires that $\lambda' \geq 0$. Given that $0 < \alpha < 1$ and $\tau > 1$, this holds strictly:

$$
\lambda' = \frac{2}{\alpha \tau} \tau^{1/\alpha}(\tau^{1/\alpha} - 1) > 0
$$

- **Complementary slackness** require that

$$
\lambda'\left(\sum_{i=1}^{n}(\boldsymbol{\theta}' . \mathbf{x}_i - y_i + 1)^\alpha - \tau n\right) = 0.
$$

Given that $\lambda' > 0$, this is equivalent to $\sum_{i=1}^{n}(\boldsymbol{\theta}' . \mathbf{x}_i - y_i + 1)^\alpha = \tau n$. Next, we have:

$$
\begin{aligned}
\sum_{i=1}^{n}(\boldsymbol{\theta}' . \mathbf{x}_i - y_i + 1)^\alpha = \tau n \quad &\Leftrightarrow \quad \sum_{i=1}^{n}(\boldsymbol{\theta}^* . \mathbf{x}_i - y_i + 1 + \tau')^\alpha = \tau n \\
&\Leftrightarrow \quad n(1 + \tau')^\alpha = n\tau \\
&\Leftrightarrow \quad \tau' = \tau^{1/\alpha} - 1
\end{aligned}
$$

- **Primal feasibility** automatically holds with equality given the complementary slackness derivation above.

∎

## C  Omitted Experimental Details

**Data sets**  For the regression task, we used the *Crime and Communities data set* [Lichman, 2013]. The data consists of 1994 observations each made up of 101 features, and it contains socio-economic, law enforcement, and crime data from the 1995 FBI UCR. Community type (e.g. urban vs. rural), average family income, and the per capita number of police officers in the community are a few examples of the explanatory variables included in the dataset. The target variable is the "per capita violent crimes" . We preprocessed the original dataset as follows: we removed the instances for which target value was unknown. Also, removed features whose values were missing for more than $80\%$ of instances. We standardized the data so that each feature has mean 0 and variance 1. We divided all target values by a constant so that labels range from 0 to 1. Furthermore, we flipped the labels to make sure higher $y$ values correspond to more desirable outcomes.

For the classification task, we used the *COMPAS dataset* originally compiled by Propublica [Larson *et al.*, 2016]. The data consists of 5278 observations each made up of the following features: intercept, severity of charge (felony or misdemeanour), number of priors, juvenile felony count, juvenile misdemeanor count, other juvenile offense count, race (African-American or white), age, gender, COMPAS scores (not included in our analysis). The target variable indicates the actual recidivism within 2 years. The data was filtered following the original study: If the COMPAS score was not issued within 30 days from the time of arrest, because of data quality reasons the instance was omitted. The recidivism flag is assumed to be -1 if no COMPAS case could be found at all. Ordinary traffic offences were removed. We standardized the non binary features to have mean 0 and variance 1. Also, we negated the labels to make sure higher $y$-values correspond to more desirable outcomes.

**Optimization program for classification**  Ideally we would like to find the optimum of the following constrained optimization problem:

$$\min_{\boldsymbol{\theta} \in \mathbb{R}^k} \quad \frac{1}{n} \sum_{i=1}^{n} \log(1 + \exp(-y_i \boldsymbol{\theta}.\mathbf{x}_i))$$

$$\text{s.t.} \quad \frac{1}{n} \sum_{i=1}^{n} u(b(y_i, sign(\boldsymbol{\theta}.\mathbf{x}_i))) \geq \tau$$

However, the sign function makes the constraint non-convex, therefore we instead solve the following:

$$\min_{\boldsymbol{\theta} \in \mathbb{R}^k} \quad \frac{1}{n} \sum_{i=1}^{n} \log(1 + \exp(-y_i \boldsymbol{\theta}.\mathbf{x}_i))$$

$$\text{s.t.} \quad \frac{1}{n} \sum_{i=1}^{n} u(b(y_i, \frac{\boldsymbol{\theta}.\mathbf{x}_i}{c})) \geq \tau,$$

$$\|\boldsymbol{\theta}\|^2 = 1$$

The constant $c$ ensures that the argument $(\frac{\boldsymbol{\theta}.\mathbf{x}_i}{c})$ of the benefit function is in $[-1, 1]$ which keeps our benefit non negative. For this particular setting we chose $c = 5$. We constrain $\boldsymbol{\theta}$ to be unit-length since otherwise one could increase the benefit without changing the classification outcome by just increasing the length of $\boldsymbol{\theta}$.

**Fairness Measures**  Suppose we have two groups $G_1$, $G_2$ and our labels for classification are in $\{-1, 1\}$. Also let

$$G^+ := \sum_{i \in G} \mathbf{1}[\hat{y}_i > y_i]$$

and similarly

$$G^- := \sum_{i \in G} \mathbf{1}[\hat{y}_i < y_i]$$

Figure 5: Accuracy and mean squared loss as the function of $\tau$ for different values of $\alpha$

- **Average violation of Dwork *et al.*'s pairwise constraints** is computed as follows:

$$\frac{2}{n(n-1)} \sum_{i=1}^{n} \sum_{j=i+1}^{n} \max\{0, |\hat{y}_i - \hat{y}_j| - d(i,j)\}.$$

At a high level, the measure is equal to the average of the amount by which each pairwise constraint is violated. For classification, we took $d(i,j)$ to be the Euclidean distance between $\mathbf{x}_i, \mathbf{x}_j$ divided by the maximum Euclidean distance between any two points in the dataset. The normalization step is performed to make sure the range of $|\hat{y}_i - \hat{y}_j|$ and $d(i,j)$ are similar. For regression, we tool $d(i,j) = |y_i - y_j|$—assuming the existence of an ideal distance metric that perfectly specifies the similarity between any two individuals' ground truth labels.

- **Demographic parity** is computed by taking the absolute difference between percentage of positive predictions across groups:

$$\left| \frac{1}{|G_1|} \sum_{i \in G_1} \mathbf{1}[\hat{y}_i = 1] - \frac{1}{|G_2|} \sum_{i \in G_2} \mathbf{1}[\hat{y}_i = 1] \right|.$$

- **Difference in false positive rate** is computed by taking the absolute difference of the false positive rates across groups:

$$|f_{fpr}(G_1) - f_{fpr}(G_2)|$$

where:

$$f_{fpr}(G) := \sum_{i \in G} \frac{\mathbf{1}[\hat{y}_i = 1 \wedge y_i = -1]}{\mathbf{1}[y_i = -1]}.$$

- **Difference in false negative rate** is computed by taking the absolute difference of the false negative rates across groups:

$$|f_{fnr}(G_1) - f_{fnr}(G_2)|$$

where:

$$f_{fnr}(G) := \sum_{i \in G} \frac{\mathbf{1}[\hat{y}_i = -1 \wedge y_i = 1]}{\mathbf{1}[y_i = 1]}.$$

- **Mean difference** is computed by taking the absolute difference of the prediction means across groups:

$$\left| \frac{1}{|G_1|} \sum_{i \in G_1} \hat{y}_i - \frac{1}{|G_2|} \sum_{i \in G_2} \hat{y}_i \right|$$

- **Positive residual difference** [Calders *et al.*, 2013] is computed by taking the absolute difference of mean positive residuals across groups:

$$\left| \frac{1}{|G_1^+|} \sum_{i \in G_1} \max\{0, (\hat{y}_i - y_i)\} - \frac{1}{|G_2^+|} \sum_{i \in G_2} \max\{0, (\hat{y}_i - y_i)\} \right|.$$

- **Negative residual difference** [Calders *et al.*, 2013] is computed by taking the absolute difference of mean negative residuals across groups:

$$\left| \frac{1}{|G_1^-|} \sum_{i \in G_1} \max\{0, (y_i - \hat{y}_i)\} - \frac{1}{|G_2^-|} \sum_{i \in G_2} \max\{0, (y_i - \hat{y}_i)\} \right|.$$

**Trade-offs with Accuracy**  Figure 5 illustrates the trade-offs between our proposed measure and prediction accuracy. As expected, imposing more restrictive fairness constraints (larger $\tau$ and smaller $\alpha$), results in higher loss of accuracy.