[Reviews · NeurIPS 2018]

Reviewer 1



This paper proposes a new measure of fairness for classification and regression problems based on welfare considerations rather than inequality considerations. This measure of fairness represents a convex constraint, making it easy to optimize for. They experimentally demonstrate the tradeoffs between this notion of fairness and previous notions. I believe this to be a pretty valuable submission. A welfare-based approach over a inequality-based approach should turn out to be very helpful in addressing all sorts of concerns with the current literature. It also provokes a number of questions to follow up on which, while disappointing that they are not addressed here, means that the community should take interest in this paper. I am not at all convinced that this is really an 'individual' notion of fairness. The requirement is to ensure only that the sum of individual utilities is sufficiently high, rather than enforcing that every utility is high (as would be the case if you forced \alpha=-\infty), or something similar. Thus I think it is not particularly appropriate to call this the "first computationally feasible mechanism for bounding individual-level (un)fairness." While on the subject of the abstract, the abstract states "our work provides both theoretical and empirical evidence suggesting that a lower-bound on our measures often leads to bounded inequality in algorithmic outcomes." Empirical evidence is certainly provided, but to what theoretical evidence does this statement refer? The connection to Atkinson's index? I ended up waiting for generalization results that didn't exist in this paper, so that could be cleared up. A few very small things: - It looks like there's overlapping notation between Pareto domination of the benefits and the welfare's total ordering (line 206) - "consistently" --> "consistent" (line 271) - Figure 2 has "accuracy" where the Dwork measure should be instead

Reviewer 2



Summary This paper proposes new family of fairness measures that incorporate social welfare and risk aversion, measures from social welfare in economics. An empirical analysis is given using the COMPAS (classification) and Crime and Communities (regression) data sets using 5 different algorithms. It demonstrates that tradeoffs between accuracy and improved fairness. Comments the abstract states "a rational, risk-averse individual who is going to be subject to algorithmic decision making and is faced with the task of choosing between several algorithmic alternatives" I found this confusing because the first part of the sentence seems to imply the individual is an end user or subject of an algroithm decision, but the 2nd part suggests that this user/subject will have the ability to choose which algorithm is used for the decision. Usually such users have no say in the choose of the algorithm. The example in the intro provides a nice motivation, but later talks about "higher welfare" without defining its meaning. It would be nice to give an informal definition when it is first referenced. The caption of Fig 2 mentions 3 metrics: Dwork measure, social welfare and Atkinson index, but the 3 charts seem to be accuracy, Atkinson, and social welfare. Fig 2c gives Dwork measures and calls it "average violation". If this is the same as 2c, it would be good to have the same label on the y axis. ----------------------------------------------- I have read the authors rebuttal and appreciate that they will clarify the explanations in the revised version of the paper.

Reviewer 3



This work approaches the question of fairness through the lens of welfare and risk appetite. That is, what are the tradeoffs between social welfare, risk adverseness, and the canonical definitions of fairness? Drawing on a rich set of references, the authors define a constrained optimization problem where loss is minimized with respect to expected utility greater than a threshold. The theoretical section followed easily although it was not clear to me what Proposition 1 contributed. The experiments are explained well, and I particularly appreciated Figures 2-4 since the work begins to engage with prior methods. I am curious if an impossibility theorem exists regarding welfare and risk and other notions of fairness. The experimental results seems to suggest this, but analysis here would greatly strengthen the paper. Minor nitpicks - Figure 1 was hard to parse since we are comparing models against each other. Suggestion to switch out legend and x axis and have the x axis be the different models and the colors correspond to groups. - Figure 2's caption refres to Dwork measure, but the y-label says accuracy. It does not appear that these are the same. I also did not understand the third sentence: "Note that for Dwork and ... for social welfare larger values reflect greater fairness." I'm assuming the left-most plot was supposed to be Dwork measure instead of accuracy? - Please proof-read again for grammar, e.g. line 267: "compare with one another according [to] accuracy" and spacing after periods (line 270).